# The Nutritional, ACE Inhibition, and Antioxidant Properties of Hydrolysate Powders Derived from Different Stages of Thai Silkworm (*Bombyx mori*)

**DOI:** 10.3390/foods14234018

**Published:** 2025-11-23

**Authors:** Artorn Anuduang, Wan Aida Wan Mustapha, Seng Joe Lim, Somchai Jomduang, Sakaewan Ounjaijean, Supakit Chaipoot, Oranit Kraseasintra, Kongsak Boonyapranai

**Affiliations:** 1Research Institute of Health Science, Chiang Mai University, Chiang Mai 50100, Thailand; a.anuduang@gmail.com (A.A.); sakaewan.o@cmu.ac.th (S.O.); 2Department of Food Sciences, Faculty of Science and Technology, Universiti Kebangsaan Malaysia, UKM, Bangi 43600, Selangor, Malaysia; wanaidawm@ukm.edu.my (W.A.W.M.); joe@ukm.edu.my (S.J.L.); 3Innovation Centre for Confectionery Technology, Faculty of Science and Technology, Universiti Kebangsaan Malaysia, UKM, Bangi 43600, Selangor, Malaysia; 4Bio Crenovation Company Limited, 353/2 Moo 9, Tambol Sanklang, Sanpatong District, Chiang Mai 50120, Thailand; admin@bio-c.co.th; 5The Traditional Food Research and Development Unit, Multidisciplinary Research Institute (MDRI), Chiang Mai University, Chiang Mai 50200, Thailand; supakit.ch@cmu.ac.th; 6Office of Research Administration, Chiang Mai University, Chiang Mai 50200, Thailand; orkraseasintra@gmail.com

**Keywords:** Thai silkworms, enzymatic hydrolysis, protease, ACE inhibitory activity, antioxidant activity

## Abstract

This study evaluated the bioactive potential of Thai silkworms (*Bombyx mori*) at three developmental stages—mature silkworm (MS), A-silking silkworm (AS), and pupae (PP)—as alternative protein sources for functional hydrolysates. Silkworm powders were hydrolyzed with Alcalase^®^ (5% *w*/*w*, 1 h, 60 °C) to obtain MS hydrolysate powder (MSHP), AS hydrolysate powder (ASHP), and PP hydrolysate powder (PPHP). AS contained the highest protein content (72.13%), followed by MS (70.20%) and PP (56.70%). Amino acid profiling revealed stage-specific and hydrolysis-dependent variations, MS was enriched in phenylalanine and histidine, AS in threonine, valine, and tyrosine, and PP in lysine, leucine, and arginine. Hydrolysates showed markedly increased amino acid levels across all samples, indicating enhanced peptide release and improved nutritional quality. The hydrolysates achieved yields of 61–64% and protein recoveries of approximately 46%. MSHP and ASHP exhibited higher degrees of hydrolysis than PPHP. Among the biological activities, MSHP demonstrated the strongest angiotensin-converting enzyme (ACE) inhibition (88.46%), whereas PPHP exhibited the greatest antioxidant capacity (DPPH, ABTS, FRAP). Overall, Alcalase^®^ hydrolysis effectively enhanced silkworm bioactivity, supporting their potential as multifunctional ingredients for functional foods and nutraceuticals targeting cardiovascular and oxidative stress-related disorders.

## 1. Introduction

Edible insects have gained significant global attention as a sustainable and nutritious protein source, aligning with international goals for food security and environmental sustainability. Insects are rich in protein and essential nutrients while requiring fewer resources than traditional livestock [1]. The practice of insect farming also generates minimal waste by upcycling organic side-streams into biomass, exemplifying the circular economy approach to responsible production and consumption. Such attributes directly support the United Nations Sustainable Development Goals—SDG 2 (Zero Hunger) through improved food and nutrition security, SDG 3 (Good Health and Well-being) via nutrient-dense dietary options and biological properties, and SDG 12 (Responsible Consumption and Production) by reducing waste and resource use in protein production [2].

Among the edible insects, the silkworm (*Bombyx mori*) stands out as a protein-rich and widely available species with dual roles in sericulture and nutrition. Silkworm larvae and pupae have long been consumed in parts of Asia, and they are emerging as nutritious food and feed ingredients due to their high protein content and bioactive compounds [3]. In fact, dried silkworm pupae consist of roughly 50–60% crude protein by weight [4]. Notably, silkworms offer not only macronutrient nutrition but also pharmacological and functional benefits. Studies have documented a range of health-promoting effects from silkworm-derived peptides and/or extracts, and silkworm pupae consumption has been reported to increase in hepatic alcohol dehydrogenase activity, enhance immunity, and lower blood pressure [5,6,7]. The prospect of harnessing such functional components from *B. mori* across its developmental stages (mature silkworm, A-silking silkworm, and pupae) thus presents an attractive strategy for developing nutraceutical ingredients while also contributing to zero-hunger and wellness objectives.

Enzymatic hydrolysates from edible insects show enhanced bioactivity over intact proteins, particularly strong ACE-inhibitory effects linked to anti-hypertensive potential. For instance, silkworm protein hydrolysates achieved nearly 100% ACE inhibition in vitro versus 50% from the unhydrolyzed form [8]. Beyond anti-hypertensive effects, insect protein hydrolysates also show potent antioxidant activity. *B. mori* pupae hydrolysates, especially those produced with Alcalase, exhibit markedly enhanced radical-scavenging capacity, attributed to peptides that neutralize free radicals or boost cellular antioxidant defenses [9]. These findings underscore the value of enzymatic hydrolysis in generating insect-derived peptides with functionalities relevant to human health, such as ACE inhibition (for cardiovascular health) and antioxidant activity (for mitigation of oxidative stress).

However, a key research gap remains, as few studies have systematically compared the biological activities of silkworm hydrolysates derived from different developmental stages. Limited information exists on how life stage affects the generation of bioactive peptides and their functional properties, especially in terms of ACE-inhibitory and antioxidant capacities. Building on this background, the present study was conceived to explore the bioactivity and functional food potential of Thai silkworm protein hydrolysates obtained from different life stages of *Bombyx mori*. This research focuses on the mature silkworm (MS), A-silking silkworm (AS), and pupae (PP) by processing their proteins with Alcalase under optimized conditions to release bioactive peptides and comparing the biochemical properties and health-related activities (especially ACE-inhibitory and antioxidant capacities) of hydrolysates from these developmental stages.

The objective of this research is to evaluate how different life stages of silkworms influence their nutraceutical potential. Specifically, it aims to identify which developmental stages are most effective for ACE-inhibitory and antioxidant activities. This study not only advances the characterization of silkworms as a sustainable source of alternative protein and functional ingredients, but also guides targeted utilization of specific life stages to maximize health-promoting properties. It is anticipated that the findings will guide future initiatives to incorporate substances produced from silkworms into functional foods or supplements, furthering the function of entomophagy in public health and food security.

## 2. Materials and Methods

### 2.1. Materials

The Thai silkworm (*Bombyx mori* L.), a hybrid variety reared by the Queen Sirikit Department of Sericulture, Ministry of Agriculture and Cooperatives (Thailand), was used in this study to examine three developmental stages.

-Mature silkworms (MSs) were obtained by rearing silkworms until they reached the mature stage, corresponding to the final instar of larval development prior to cocoon spinning. The duration of this stage is approximately 20–25 days, depending on environmental conditions. MSs are characterized by a translucent yellowish coloration.-Silking silkworms (ASs) represent silkworms in the mature phase, which typically generate cocoons naturally. However, specific techniques can stimulate silk sheet production instead of cocoons, thereby preventing metamorphosis into pupae.-Pupae (PP) were obtained by allowing mature silkworms to spin cocoons for 3–5 days, during which they underwent metamorphosis into pupae enclosed within the cocoons.

### 2.2. Silkworms Hydrolysis Preparation

Powders from different developmental stages of silkworms were prepared. A total of 2 kg of MSs, ASs, and PP were processed by boiling for 5 min, followed by vacuum microwave drying for 2.5 h at 4800 kW magnetron power. The dried MS, AS, and PP samples were subsequently ground into powder using a high-speed blender operating at 25,000 rpm (Nanotech, Beijing, China), as described in our previous study [10]. MS, AS and PP powder were kept at −18 °C in freezer for further study.

The enzymatic hydrolysis conditions, including enzyme type, concentration, temperature, pH, and reaction time, were selected according to the method reported by Anuduang et al. [10] which achieved the highest ACE-inhibitory activity and yield among the tested conditions. Powders from mature silkworms (MSP), A-silking silkworms (ASP), and pupae (PPP) were hydrolyzed using Alcalase^®^ 2.4 L (2.54 AU-A/g activity) (Novozymes, Bagsværd, Denmark). Briefly, 5 g of each powder was hydrolyzed with Alcalase at 5% (*w*/*w*) of the protein content in 50 mL of 0.1 mM phosphate buffer (pH 8.0) and incubated at 60 °C for 1 h. After hydrolysis, the samples were heated in a water bath at 95 °C for 10 min and then centrifuged at 1008× *g* for 30 min. The supernatants were collected and freeze-dried (Christ, Alpha 1-4 LSC plus, Osterode am Harz, Germany). The resulting hydrolysates MSHP, ASHP, and PPHP were stored at −18 °C for further analysis.

The yields of each silkworm powder were calculated based on the following formula:$$\mathrm{yield}\;(\%)\;=\;\frac{weight\;of\;silkworm\;hydrolysate\;powder\;(g)}{weight\;of\;silkworm\;powder\;(g)}\times 100$$

### 2.3. Proximate Analysis

Proximate analysis of mature silkworm powder (MSP), A-silking silkworm (ASP), and pupae powder (PPP) were analyzed using the protocols of the Association of the Official Analytical Chemists (AOAC, 2012 [11]). Moisture content (AOAC 934.01) was determined by drying approximately 2 g of MSP, ASP, and PPP at 105 °C until a constant weight was achieved. Subsequently, the moisture content was calculated. Ash content (AOAC 923.03, 2012) was determined by incinerating approximately 2–3 g of MSP, ASP, and PPP at 550 °C in a combustion chamber (Neytech, Vulcan 3-550, Plymouth, MN, USA)until grey ash formed, and calculating the ash percentage from the residue weight. In terms of total fat content (AOAC 991.36), this was determined using 3 g of MSP, ASP, and PPP via Soxhlet extraction (BÜCHI, E-800, Flawil, Switzerland)with petroleum ether (boiling point 40–60 °C, ≥99% purity, analytical grade; Sigma-Aldrich, St. Louis, MO, USA) as the solvent, and quantified gravimetrically. Crude protein content was determined by nitrogen analysis using a combustion instrument (LECO Nitrogen and Protein Analyzer; FP828, St. Joseph, MI, USA)). Approximately 0.20 ± 0.05 g of silkworm samples, together with EDTA as a reference, were accurately weighed. MSP, ASP, PPP, MSHP, ASHP, and PPHP were carefully wrapped in tin foil and then loaded into the protein analyzer. The protein content in the samples was determined by applying a relevant equation. Total protein content (%) = %Nitrogen × 6.25 and carbohydrate was obtained by the formula of 100—(the sum of moisture, protein, fat, and ash). It should be noted that under these conditions, crude protein represents nitrogen-based protein and may include contributions from non-protein nitrogenous components such as chitin.

### 2.4. Degree of Hydrolysis

The degree of hydrolysis of MSHP, ASHP, and PPHP was determined using The O-Phthaldialdehyde (OPA) method following the method of Anuduang et al. [10]. MSHP, ASHP, and PPHP solution (0.1 g/mL) was combined with 2 mL of the OPA reagent and incubated for 2 min at room temperature. The absorbance of the resulting mixture was measured at 340 nm using a microplate reader (SPECTROstar Nano, BMG LABTECH GmbH, Ortenberg, Germany). For the standard, approximately 50 mg of α-serine (Sigma-Aldrich, Darmstadt, Germany) was dissolved in 500 mL of deionized water (DI).$$\mathrm{Serine}\text{-}{\mathrm{NH}}_{2}=[\frac{\left(OD\;sample-OD\;blank\right)}{\left(OD\;standard-OD\;blank\right)}]\;\times \;0.9156\;\mathrm{meqv}/L\;\times \;0.1\;\times \;(\frac{100}{\left(x\times P\right)})$$ where x = gram of hydrolysate sample, and *P* = % protein content in sample.$$\mathrm{Determination}\;\mathrm{of}\;h=(\frac{SerineNH2-\beta }{\alpha })$$
where *β* = 0.4 and *α* = 1.$$\mathrm{Calculation}\;\mathrm{degree}\;\mathrm{of}\;\mathrm{hydrolysis}=\left(\frac{h}{htot}\right)\times 100$$
where *h_tot_* = 7.6.

### 2.5. Free Amino Acids

The amino-acid composition of MSP, ASP, PPP and their hydrolysates (MSHP, ASHP, PPHP) was determined by HPLC with post-column derivatization following the Shimadzu Na-type protocol [12]. Analyses were performed on a Shimadzu Prominence HPLC equipped with a Prominence RF-20A fluorescence detector (Shimadzu, Kyoto, Japan) and a Shim-pack Amino-Na column (100 mm × 6.0 mm i.d., 5 µm, P/N 228-18837-91, Shimadzu, Kyoto, Japan). A three-solvent gradient was used: Mobile phase A was a sodium citrate buffer (pH 3.23) containing 19.6 g trisodium citrate dihydrate, 70 mL absolute ethanol, 14.3 mL 70% perchloric acid, and 0.1 mL octanoic acid, made up to 1 L with water; mobile phase B was a sodium citrate/borate buffer (pH 10.00) prepared from 58.8 g trisodium citrate dihydrate, 12.4 g boric acid, and 5.2 g sodium hydroxide, diluted to 1 L with water; and mobile phase C was 0.2 M sodium hydroxide. After chromatographic separation, amino acids were derivatised post-column with o-phthalaldehyde (OPA) and N-acetylcysteine in a reaction coil and detected fluorimetrically by the RF-20A detector. Operating conditions were: column oven 60 °C, flow rate 0.40 mL min^−1^, injection volume 10 µL, and a total run time of 65 min. Samples were prepared by dissolving each extract in 0.1 M HCl adjusted to pH 2.2, filtering through a 0.45 µm syringe filter, and injecting 10 µL. Quantification used external calibration against mixed amino-acid standards; concentrations were calculated from calibration curves and expressed as mg per 100 g dry weight.

### 2.6. Determination of the Angiotensin-Converting Enzyme (ACE) Inhibitory

The angiotensin-converting enzyme (ACE) inhibitory activity of MSP, ASP, PPP and their hydrolysates (MSHP, ASHP, PPHP) was determined using a colorimetric assay with the ACE Kit-WST (Dojindo Inc., Kumamoto, Japan), as described in Anuduang et al. [10]. Briefly, 20 µL of each sample (2 mg/mL), Enalapril (positive control, 2 mg/mL), and deionized water (control and blank) was added to a 96-well plate. Then, 20 µL of substrate buffer was added to all wells, followed by 20 µL of enzyme working solution to the sample, positive control, and control wells (deionized water was added to the blank). The plate was incubated at 37 °C for 1 h, after which 200 µL of indicator working solution was added, followed by a further incubation at 37 °C for 10 min. Absorbance was measured at 450 nm using a microplate reader (SPECTROstar Nano, BMGLABTECH, Germany).$$\mathrm{ACE}\;\mathrm{inhibition}\;(\%)=(\frac{Absorbance\;of\;control-Absorbance\;of\;sample}{absorbance\;of\;control-Absorbance\;of\;blank})\times 100$$

### 2.7. Silkworm Hydrolysate Fractions

The mature silkworm hydrolysate (MSH) was subjected to fractionation through ultrafiltration employing Amicon Ultra-15 centrifugal filter units (Merck Millipore, Melbourne, Australia). To outline the procedure briefly, 10 mL of MSH was loaded into separate Amicon Ultra-15 units equipped with Ultracel-10 membranes (with a Molecular Weight Cutoff (MWCO) of 10 kDa). Subsequently, these were centrifuged (Hettich, Tuttlingen, Germany) at 5000 g and maintained for 30 min at room temperature, yielding a retentate fraction with a molecular weight greater than 10 kDa. The permeate from the above step was then introduced into a separate Amicon Ultra-15 unit equipped with Ultracel-3 membranes (with a MWCO of 3 kDa). Following another centrifugation step at 5000 *g* for 30 min at room temperature, two distinct fractions were obtained. The retentate constituted a molecular weight fraction ranging from 3–10 kDa, while the permeate consisted of molecules with a molecular weight less than 3 kDa.

### 2.8. Total Phenolic Content

Total phenolic compounds (TPCs) in MSHP, ASHP, and PPHP were determined by the Folin–Ciocalteu method. Each sample (2 mg in 1 mL of deionized water) was mixed with 150 µL of 10% (*v*/*v*) Folin–Ciocalteu reagent (Merck, Darmstadt, Germany) and left at room temperature for 3 min. Then, 100 µL of 7.5% (*w*/*v*) Na_2_CO_3_ (Sigma, Darmstadt, Germany) was added, and the mixture was incubated in the dark for 30 min. Absorbance was measured at 760 nm using a microplate reader (SPECTROstar Nano, BMG LABTECH, Germany). Gallic acid (concentration; 0.025, 0.05, 0.1, 0.2, 0.3, and 0.4 mg/mL) was used as the standard, and TPCs were expressed as mg gallic acid equivalents (GAE)/mL.

### 2.9. Antioxidant Activity Assays

#### 2.9.1. The 2,2-Diphenyl-1-picrylhydrazyl (DPPH) Free Radical Scavenging Assay

The DPPH free radical scavenging activity of MSHP, ASHP, and PPHP was determined as described in Phongthai et al. [13], with slight modification. Briefly, 10 µL of each sample (20, 10, 5, and 2.5 mg/mL), Trolox (0.20, 0.15, 0.10, 0.05, and 0.025 mg/mL; standard), or blank was mixed with 190 µL of DPPH solution in a 96-well plate. The mixtures were incubated in the dark at room temperature for 30 min. Absorbance was measured at 517 nm using a microplate reader (SPECTROstar Nano, BMGLABTECH, Germany). All measurements were performed in triplicate, and results were expressed as IC_50_ values (mg/mL).

#### 2.9.2. The 2,2′-Azino-bis(3-ethylbenzothiazoline-6-sulfonic Acid (ABTS) Assay

The ABTS radical scavenging activity of MSHP, ASHP, and PPHP was determined as described in Phongthai et al. [13], with slight modification. Briefly, 10 µL of each sample (20, 10, 5, and 2.5 mg/mL) was mixed with 290 µL of ABTS stock solution and incubated in the dark at room temperature for 10 min. The absorbance was measured at 734 nm using a microplate reader (SPECTROstar Nano, BMGLABTECH, Germany). Deionized water served as the negative control. Results were expressed as IC_50_ values (mg/mL).

#### 2.9.3. Ferric Reducing Antioxidant Power (FRAP) Assay

The ferric reducing antioxidant power (FRAP) of MSHP, ASHP, and PPHP was determined as described in Phongthai et al. [13], with slight modification. The FRAP working reagent was freshly prepared by mixing 300 mM acetate buffer (pH 3.6), 10 mM TPTZ (2,4,6-tris[2-pyridyl]-s-triazine) in 40 mM HCl, and 20 mM FeCl_3_·6H_2_O in a 10:1:1 ratio. The reagent was light orange in color. Then, 15 µL of each sample (2 mg/mL) was added to 285 µL of FRAP reagent and incubated at room temperature for 10 min. Absorbance was measured at 595 nm using a microplate reader (SPECTROstar Nano, BMGLABTECH, Germany). A calibration curve of FeSO_4_ was used to estimate antioxidant capacity, and results were expressed as mg Fe^2+^ equivalents per mL. Deionized water served as the negative control.

### 2.10. Statistical Analysis

The results are expressed as the mean ± standard deviation (SD) of three independent experiments. Statistical analyses were performed using one-way ANOVA with SPSS version 20 (IBM Corp., Armonk, NY, USA). Prior to analysis, the assumptions of normality and homogeneity of variance were verified using the Shapiro–Wilk and Levene’s tests, respectively. Differences among treatments for amino acid profile, yield, protein content, degree of hydrolysis, total phenolics, antioxidant activities (DPPH, ABTS, and FRAP), and ACE inhibition were analyzed using a completely randomized design (CRD). When significant effects were observed (*p* < 0.05), mean separations were conducted using Duncan’s multiple range test.

## 3. Results

### 3.1. Proximate Analysis of Different Stage Silkworm Hydrolysate Powder

The different types of silkworm powder, including MSP, ASP, and PPP, were used as raw materials for the hydrolysis process. Upon subjecting MSP, ASP, and PPP to proximate analysis after the drying process and grinding, it was found that under the same conditions, all samples achieved a final moisture content below 3%, ranging from 1.54% to 2.51% (Table 1). Protein content differed significantly among the samples. ASP contained the highest protein level (72.13 ± 0.70%), which was significantly greater than that of MSP (70.20 ± 0.52%) (*p* < 0.05). Both ASP and MSP had markedly higher protein contents than PPP (56.70 ± 1.13%) (*p* < 0.05). For fat content, PPP showed the highest proportion (30.44 ± 0.89%), followed by ASP (21.10 ± 0.44%) and MSP (17.81 ± 0.19%). All values were statistically different from one another (*p* < 0.05). In terms of ash content, this also varied significantly, with PPP containing the highest amount (5.06 ± 0.02%), MSP showing an intermediate level (4.14 ± 0.04%), and ASP the lowest (3.65 ± 0.12%) (*p* < 0.05). Carbohydrate content was significantly greater in PPP (6.03 ± 1.50%) compared to MSP (5.34 ± 0.38%) and ASP (1.50 ± 0.96%) (*p* < 0.05). Additionally, MSP had significantly a higher carbohydrate content than ASP (*p* < 0.05).

### 3.2. Free Amino Acid of Different Stages of Silkworm Powder

The amino-acid composition differed significantly among MSP, ASP and PPP (*p* < 0.05) (Table 2). For essential amino acids, PPP contained the highest histidine (319.69 ± 1.57 mg/100 g), isoleucine (51.36 ± 0.00 mg/100 g), leucine (61.80 ± 0.19 mg/100 g), lysine (142.94 ± 0.36 mg/100 g), and arginine (68.25 ± 1.12 mg/100 g). ASP showed significantly higher threonine (92.32 ± 0.49 mg/100 g) and valine (95.48 ± 0.08 mg/100 g) than the other powders, whereas MSP had the greatest phenylalanine (363.08 ± 8.48 mg/100 g) and methionine (27.51 ± 0.30 mg/100 g). For non-essential amino acids, MSP was highest for glutamic acid (320.72 ± 1.39 mg/100 g) and alanine (150.44 ± 0.51 mg/100 g); ASP was highest for serine (183.97 ± 0.02 mg/100 g) and tyrosine (171.22 ± 8.78 mg/100 g); and PPP contained the most proline (80.91 ± 0.10 mg/100 g). Glycine varied less across powders (ASP 131.58 ± 0.02, MSP 125.24 ± 2.43, PPP 117.87 ± 0.17 mg/100 g), with ASP significantly higher than PPP, while MSP was intermediate.

Hydrolysis markedly altered the profiles (different upper-case letters, *p* < 0.05). Among the hydrolysates, PPHP had the highest histidine (976.75 ± 0.71 mg/100 g), isoleucine (77.36 ± 0.00 mg/100 g), leucine (152.28 ± 0.12 mg/100 g), and lysine (185.94 ± 0.84 mg/100 g); ASHP was highest for threonine (129.14 ± 0.14 mg/100 g), valine (174.82 ± 0.38 mg/100 g), serine (300.00 ± 0.79 mg/100 g), and tyrosine (246.49 ± 0.57 mg/100 g); and MSHP was highest for phenylalanine (1234.61 ± 3.27 mg/100 g), alanine (225.18 ± 0.75 mg/100 g), cystine (181.03 ± 0.95 mg/100 g), and glutamic acid (489.45 ± 2.49 mg/100 g). Glycine also increased after hydrolysis (MSHP 150.02 ± 0.02; ASHP 157.20 ± 0.25; PPHP 143.43 ± 0.28 mg/100 g). Two notable exceptions were aspartic acid, which increased markedly in ASHP (6.73 ±0.01 to 38.34 ± 0.14) and PPHP (3.30 ± 0.03 to 8.30 ± 0.11 mg/100 g) but remained reduced in MSHP (5.74 ±0.14 to 2.27 ± 0.06 mg/100 g), and tyrosine and aspartic acid in the MSP pair, which decreased after hydrolysis (5.74 ±0.14 to 2.27 ± 0.06 mg/100 g and 56.33 ± 0.83 to 48.22 ± 0.47 mg/100 g, respectively). Overall, both silkworm stage and hydrolysis significantly shaped the amino-acid profiles, with hydrolysis generally increasing the free amino-acid levels.

### 3.3. Yield, Protein Content, and Degree of Hydrolysis of Different Stage Silkworm Hydrolysate Powder

The yield, protein content, and degree of hydrolysis of silkworm hydrolysate powders, which were hydrolyzed with Alcalase^®^ 2.4 L, were evaluated based on different developmental stages of the silkworm. Under similar hydrolysis conditions, the results indicate that the developmental stage of the silkworm did not significantly affect the hydrolysate powder yield (ranging from 60.97% to 63.66%) and protein content (ranging from 45.97% to 46.97%) (*p* > 0.05) (Figure 1A,B). However, in terms of the degree of hydrolysis (DH), both MSHP and ASHP exhibited similar DH values of 28.92 ± 0.13 and 28.35 ± 0.48, respectively (*p* > 0.05), which were significantly higher than that of the PPHP at 26.64 ± 0.20 (*p* < 0.05) (Figure 1C).

### 3.4. Inhibition of Angiotensin Converting Enzyme of Silkworm Powder and Silkworm Hydrolysate Powder from Different Stages of Silkworm

All types of silkworm powders and silkworm hydrolysates were evaluated for their angiotensin-converting enzyme (ACE) inhibitory activity to determine which form exerted the strongest effect. The ACE-inhibition assay was performed using the ACE Kit-WST, which employs rabbit lung–derived somatic ACE containing both N- and C-terminal catalytic domains and the substrate hippuryl-histidyl-leucine (HHL). This method therefore measures the total ACE activity, without distinguishing between specific subunits, and reflects the overall inhibitory potential.

The positive control, enalapril, exhibited nearly 100% ACE inhibition (Figure 2), serving as a benchmark. Among the silkworm hydrolysates, MSHP demonstrated the highest ACE-inhibition (88.46 ± 1.43%), which was significantly greater than that of ASHP (83.01 ± 1.22%) and PPHP (85.62 ± 1.37%) (*p* < 0.05). In contrast, the non-hydrolyzed silkworm powders of all developmental stages exhibited markedly lower inhibitory activity (approximately 50%), indicating that enzymatic hydrolysis substantially enhanced bioactivity. Statistical analysis revealed that MSHP was significantly higher than ASHP but not different from PPHP, while all hydrolysates remained inferior to enalapril. Taken together, these results suggest that hydrolysis effectively releases bioactive peptides capable of inhibiting ACE, with MSHP showing the greatest potency. However, as the assay measures the total ACE inhibition from somatic ACE, further studies using domain-specific or recombinant ACE isoforms would be necessary to determine which catalytic site (N- or C-domain) is preferentially inhibited.

MSHP, which showed the highest ACE-inhibition, was fractionated into three molecular weight ranges (≥10, 3–10, and ≤3 kDa) to clarify the molecular size distribution of ACE-inhibitory peptides, The ≤3 kDa fraction exhibited the strongest ACE inhibition (97.01 ± 0.12%), comparable to enalapril (97.96 ± 0.15%) and was significantly higher (*p* < 0.05) than the ≥10 kDa (94.64 ± 1.29%) and 3–10 kDa (94.7 ± 0.94%) fractions. These findings indicate that ACE-inhibitory activity is primarily attributed to low-molecular-weight peptides (<3 kDa) (Appendix A). Consistently, molecular weight profiling revealed that more than half of the peptides in MSHP (52.81 ± 0.95%) were <3 kDa, while only 25.50 ± 0.45% and 20.60 ± 2.00% were ≥10 kDa and 3–10 kDa, respectively. This predominance of small peptides likely facilitates efficient access to the ACE catalytic site, supporting their higher inhibitory potency.

### 3.5. Total Phenolic Compounds and Antioxidant Activities of Silkworm Hydrolysate Powder from Different Stages of Silkworm

The assessment of total phenolic compounds (TPC) and antioxidant activities including DPPH, ABTS, and FRAP in silkworm hydrolysate powder from different developmental stages revealed that PPHP exhibited the highest levels in all assays. PPHP showed the most potent TPC (26.47 ± 0.62 mg Gallic acid/mL) (Figure 3A), as well as the strongest antioxidant activities DPPH (IC_50_ = 10.20 ± 0.32 mg Trolox/mL) (Figure 3B), ABTS (IC_50_ = 1.09 ± 0.04 mg Trolox/mL) (Figure 3C), and FRAP (24.02 ± 1.07 mg Fe^2+^/mL) (Figure 3D), with significant differences (*p* < 0.05) compared to MSHP and ASHP.

## 4. Discussion

### 4.1. Proximate Analysis of Different Stage Silkworm Hydrolysate Powder

The higher protein content in ASP and MSP can be attributed to the presence of silk glands within their bodies. A recent study confirmed that the silkworm silk gland synthesizes and secretes fibroin and sericin, which are the two major protein components of cocoon silk [14]. Notably, the total protein content of PP was significantly lower compared with ASP and MSP, which can be attributed to the transformation of the silk gland into the cocoon. The protein content observed in PP is consistent with previous research findings [15]. In terms of fat composition, PP exhibited a significantly higher fat content (30.44% ± 0.89%) compared with MSP and ASP. This increase is likely attributable to the reduction in protein proportion during pupal development, which consequently leads to a relative increase in fat content. This fat content in PP falls within the range of 21–38% according to David-Birman et al. and Banday et al. [15,16]. Regarding ash content, all types of silkworm samples fell within the range of 3.65% to 5.06%, consistent with previous studies [15,17]. MS, AS, and PP had a carbohydrate content ranging from 1.50% to 6.03%. Top of Form

### 4.2. Amino Acid Profile of Different Stages of Silkworm Powder

In the non-hydrolyzed powders, this method quantifies total amino acids released by acid hydrolysis, whereas in MSHP, ASHP, and PPHP, it measures free amino acids and small peptides generated during enzymatic hydrolysis. Distinguishing between these analytical bases is essential: studies reporting free amino acids should be compared only with our hydrolysates, while those reporting total amino acids should be compared with the untreated powders (MSP, ASP, PPP). This approach ensures methodological consistency and avoids misleading cross-study comparisons.

In terms of different stages of silkworm powders, the observed shifts in amino acid (AA) profiles from MSP through ASP to PP reflect known aspects of lepidopteran metamorphosis. Larval silk glands produce fibroin and sericin, which are rich in glycine, alanine, serine, and threonine; for instance, fibroin is composed of approximately 45% glycine and 30% alanine [18]. In our data, alanine, glycine, and serine were abundant in the MSP and ASP stages but significantly lower in pupae, which aligns with their incorporation into silk and the degradation of sericin-rich glandular tissues. In contrast, levels of basic and structural amino acids lysine, arginine, valine, isoleucine, and proline, increased in the pupal stage. For example, pupae contained about 142 mg/g of lysine and 68 mg/g of arginine, surpassing the levels observed during larval stages. This likely reflects the mobilization of storage proteins (e.g., hexamerins) and the synthesis of cuticular and muscle proteins during pupation [19].

These stage-dependent differences are interpreted in the context of total amino-acid measurements, which aligns with the analytical basis commonly used in studies of silk fibroin, sericin, and pupal storage proteins. Considering these values within a total-AA framework allows for a reliable comparison between our MSP, ASP, and PPP samples and previously reported developmental profiles.

Beyond these direct comparisons, the patterns observed across stages also reflect broader biochemical features of insect development. Many insects characteristically exhibit high levels of glutamate, alanine, and aspartate but consistently low cysteine concentrations [20]. Our silkworm samples followed this trend: pupae showed elevated glutamic acid and alanine but minimal cysteine. The decrease in methionine from MSP to ASP further suggests active utilization of sulfur-containing amino acids during silk synthesis. The transient rise in tyrosine and serine during the ASP stage may indicate the accumulation of intermediary degradation products that are later recycled during pupal tissue remodeling. Collectively, these shifts illustrate the coordinated metabolic transitions associated with silk-gland regression, larval tissue breakdown, and the reallocation of nitrogen resources toward pupal structural development [18].

Enzymatic hydrolysis of silkworm protein significantly increased the levels of measurable free amino acids (AAs) in all three protein powders. In our analysis, most amino acids rose several-fold post-hydrolysis. For instance, phenylalanine in MSHP increased from 363 to 1235 mg/100 g, while arginine in PPHP rose from 68 to 95 mg/100 g. These increases align with findings by Anootthato et al. [21], who reported that the Alcalase hydrolysis of Bombyx mori pupae raised free or readily releasable AAs from ~9.4 to 42–62 mg/100 g; thus, their data are more comparable to our enzymatically hydrolyzed samples (MSHP, ASHP, PPHP) than to the intact powders. This enzymatic cleavage process liberates peptide-bound AAs and generates low-molecular-weight peptides, thereby improving solubility and nutritional accessibility. Notably, essential AAs (EAA) were especially elevated. For instance, PPHP had the highest lysine, isoleucine, and leucine among the three, whereas MSHP was uniquely rich in phenylalanine. ASHP showed the highest threonine, serine, and tyrosine. Overall, each stage yielded a balanced EAA set, consistent with reports that silkworm proteins contain all 18 AAs (eight essential meeting WHO standards) [22]. In short, hydrolysis exposed a larger pool of both essential and non-essential AAs, enhancing nutritional value. This pattern mirrors other insect studies. For example, Nuruk (microbial) hydrolysis of mealworm gave ~1.7× the free AAs of Alcalase treatment [23], and this comparison again specifically concerns the free AA fraction rather than total AAs.

Considered together, the stage-dependent differences in amino-acid composition among MSP, ASP, and PPP reflect key physiological transitions during metamorphosis, including silk-gland degradation and the redistribution of protein resources. Enzymatic hydrolysis further increases amino-acid availability by liberating free AAs and small peptides. Distinguishing between total and free amino acids is therefore essential for accurate interpretation, as it ensures valid comparison with previous studies and clarifies that the substantial increases observed after hydrolysis result from peptide cleavage rather than developmental changes alone.

### 4.3. Yield, Protein Content, and Degree of Hydrolysis of Different Stage Silkworm Hydrolysate Powder

The hydrolysate yield of MSHP in this study was slightly higher than that reported in our previous work [10]. This variation in results could potentially be attributed to differences in the rearing batches of mature silkworms. Furthermore, silkworm hydrolysate powders, including MSHP, ASHP, and PPHP, which were hydrolyzed using the commercial protease Alcalase^®^ 2.4 L, demonstrated a higher percentage yield compared to silkworm pupal proteins extracted via freeze–thaw-assisted extraction, ultrasound-assisted extraction, and microwave-assisted extraction [24].

Alcalase has been widely used to hydrolyze silkworm proteins, yielding a high recovery of protein in hydrolysate form. In one recent study on silkworm pupae, sequential processing (defatting, extraction, and Alcalase hydrolysis) retained about 74.7% of the initial protein after enzymatic hydrolysis compared with ~84.2% after extraction alone [25]. This suggests that most of the extracted protein is preserved during the enzymatic step, with only a moderate additional loss occurring during hydrolysis. The resulting silkworm pupae hydrolysate powder was still rich in protein (approximately 75% protein content, versus 84% in the defatted silkworm material before hydrolysis) [21]. Such high protein recovery underscores Alcalase’s efficiency in liberating silkworm proteins into soluble form. Notably, the A-silking silkworms are themselves highly proteinaceous, with up to about 50–56% protein on a dry weight basis [26], providing an excellent substrate for producing protein hydrolysates. By applying Alcalase to these silkworm byproducts, researchers can extract and concentrate this protein into a refined powder. In practical terms, producing a silkworm protein hydrolysate is quite efficient; our previous study on mature silkworm larvae (final-instar larvae before cocooning) noted that a crude hydrolysate could be prepared simply by defatting, grinding, and one-step Alcalase digestion of the whole insect without any complex fractionation, making it a low-cost process [10].

Alcalase tends to produce a much higher DH than other commonly used proteases. For example, earlier comparative work showed that Alcalase achieved significantly greater protein breakdown in silkworm pupae than the milder protease Neutrase [21]. SDS-PAGE analysis of silkworm pupae proteins during Alcalase hydrolysis confirmed the enzyme’s strong proteolytic activity. After treatment, most of the protein was converted into peptides smaller than 25 kDa, with a large proportion even falling below 10 kDa, demonstrating the effectiveness of Alcalase in breaking down silkworm proteins into low-molecular-weight peptides [25]. A high DH is advantageous because it can improve protein solubility and bioactivity. However, the specific DH obtained will depend on enzyme concentration, time, and conditions. For instance, a recent optimization with house cricket proteins (using Alcalase Novozymes, Bagsværd, Denmark) demonstrated that DH could be tuned by altering the enzyme-to-substrate ratio and incubation time, reaching an optimal point that balanced extensive hydrolysis with desired bioactivity [27]. In silkworms, researchers typically select conditions (5% Alcalase, 50–60 °C, 1–5 h) that yield a high DH while preserving functionality of the peptides. The net result is that Alcalase treatment produces a silkworm protein hydrolysate with a high degree of hydrolysis, effectively unlocking a broad range of peptides from the initial silkworm proteins.

Alcalase enzymatic hydrolysis has proven highly effective in the valorization of silkworm proteins. This method produces a high-yield silkworm hydrolysate powder with elevated protein content and achieves a substantial degree of hydrolysis, efficiently breaking down silkworm proteins into smaller peptides with potential functionality or bioactivity. This enzymatic approach remains effective across different silkworm developmental stages, from mature larvae to pupae, with only minor variations in yield and composition due to natural biological differences. The consistency of performance highlights the robustness of Alcalase as a hydrolytic agent in insect protein processing.

### 4.4. Inhibition of Angiotensin Converting Enzyme of Silkworm Powder and Silkworm Hydrolysate Powder from Different Stages of Silkworm

The mechanisms by which insect-derived peptides inhibit ACE are broadly similar to those of other peptide-based ACE inhibitors. They act mainly by binding to the active-site catalytic domain of ACE through multiple hydrogen bonds, thereby blocking the conversion of angiotensin I to angiotensin II [28]. For insect-derived peptides, inhibitory potency is strongly sequence-dependent: high-activity peptides typically contain hydrophobic or aromatic residues, such as phenylalanine or tyrosine, at the C-terminus, often accompanied by proline or other bulky residues that enhance binding affinity and increase resistance to enzymatic degradation. In *Acheta domesticus* (house cricket) hydrolysates, potent ACE-inhibitory peptides such as AVQPCF, PIVCF, and CAIAW share a C-terminal phenylalanine or tryptophan, a feature that promotes strong interactions with the ACE active site and enhances inhibitory efficacy [29]. Similarly, some peptides from silkworm hydrolysates possess negatively charged C-terminal residues (e.g., glutamic or aspartic acid) combined with N-terminal hydrophobic amino acids, a structural pattern associated with improved ACE binding affinity and greater blood-pressure-lowering potential [30].

Our findings suggest that MSHP contains bioactive constituents capable of effectively inhibiting angiotensin-converting enzyme (ACE). All silkworm-derived hydrolysate powders showed considerable ACE-inhibitory activity, indicating the presence of peptides with functional ACE-inhibiting properties [7,31]. This effect may be related to the silkworm’s exclusive mulberry leaf diet, which enriches aromatic and hydrophobic amino acids such as tryptophan, tyrosine, isoleucine, and phenylalanine, all known for their ACE-inhibitory potential [28,32]. Consistent with this, mulberry leaf extracts obtained by aqueous and ethanol-based methods have been reported to lower blood pressure and exert antihypertensive effects in animal models [33,34]. In our hydrolysates, hydrophobic and aromatic residues were enriched: MSHP was characterized by elevated phenylalanine, methionine, and alanine; ASHP by tyrosine and valine; and PPHP by leucine, isoleucine, and proline. Notably, these residues, particularly phenylalanine, tyrosine, isoleucine, leucine, and valine, are widely associated with ACE-inhibitory potential [28,32].

Recent literature further supports silkworm hydrolysates as a promising source of antihypertensive peptides. Most work to date has focused on *Bombyx mori* pupae, where enzymatic hydrolysis yields short peptide sequences with strong ACE-inhibitory effects; for example, Trp-Trp, Gly-Asn-Pro-Trp-Trp, and Lys-His-Val show high in vitro inhibitory activity [7]. In our study, MSHP achieved ~88% ACE inhibition, significantly exceeding ASHP and PPHP, which is consistent with emerging evidence that peptides from more developed silkworm stages may exhibit greater bioactivity. Wang et al. [35] further noted that, although pupal peptides have been widely investigated, mature silkworm hydrolysates remain underexplored despite their potential. The heightened activity observed in MSHP may reflect age-related biochemical changes that promote the accumulation or release of functional peptides, a hypothesis that warrants further targeted characterization.

The superior ACE-inhibitory activity of MSHP may stem from differences in protein conformation and enzymatic accessibility in mature silkworms compared with after-silking or pupal stages. During maturation, extensive proteomic remodeling increases the synthesis of storage and structural proteins enriched in hydrophobic and aromatic residues [3,14]. The amino-acid profile (Table 2) supports this, showing that MSHP contained higher levels of phenylalanine, leucine, isoleucine, valine, and methionine than the other hydrolysates. These residues strengthen ACE binding through hydrophobic interactions within the catalytic pocket, contributing to the higher inhibitory potency observed in MSHP [29,32]. In addition, partial protein cross-linking or oxidation during maturation may promote the release of shorter, more active peptides upon hydrolysis [6]. The abundance of methionine- and phenylalanine-rich sequences may also enhance radical-scavenging and metal-chelating capacity, indirectly stabilizing peptide–ACE interactions [36,37]. Collectively, these results suggest that maturation-related biochemical changes increase both the yield and potency of ACE-inhibitory peptides in MSHP.

The accumulating evidence for ACE-inhibitory peptides from silkworms points toward exciting applications in the food and health industries. These peptides offer a natural alternative to synthetic ACE inhibitor drugs, potentially without the side effects (e.g., cough, hyperkalemia) associated with long-term medication [38]. Protein hydrolysates derived from any developmental stage of the silkworm can be formulated into targeted functional foods such as protein bars, savory seasonings, and ready-to-drink beverages to deliver standardized doses of ACE-inhibitory bioactive peptides, offering a credible dietary strategy for blood-pressure management.

### 4.5. Total Phenolic Compounds and Antioxidant Activities of Silkworm Hydrolysate Powder from Different Stages of Silkworm

Although TPCs are commonly associated with plant-derived matrices, their presence in animal-based materials such as *Bombyx mori* is well documented. Because silkworms feed exclusively on polyphenol-rich mulberry leaves, these compounds are absorbed through the midgut and subsequently metabolized via conjugation pathways such as sulfation and glycosylation [39]. Comprehensive profiling has confirmed the presence of phenolic acids and flavonoids including ferulic acid, cinnamic acid, catechin, and epicatechin in *B. mori* pupae, reflecting their dietary origin and tissue accumulation [40]. Quantifying TPCs in silkworm hydrolysates is therefore biologically relevant, as phenolics derived from the mulberry diet, together with endogenously modified forms, can act synergistically with antioxidant peptides to scavenge free radicals and chelate transition metals. This combined contribution of peptide- and phenolic-based antioxidants likely underlies the superior antioxidant performance observed in PPHP.

Recent studies have shown that the enzymatic hydrolysis of silkworm pupae proteins significantly enhances their in vitro antioxidant activities. Hydrolysates derived from *Bombyx mori* pupae using various proteases demonstrated strong radical scavenging capacities, as observed through assays such as DPPH, ABTS, and FRAP [9]. These findings consistently demonstrate that protein hydrolysis not only improves antioxidant potential compared to unhydrolyzed silkworm protein, but also produces peptides with enhanced antioxidant properties through radical scavenging and reducing activities.

In this study, PPHP exhibited the most potent antioxidant activities among the tested samples. Specifically, PPHP showed the lowest IC_50_ values for DPPH and ABTS scavenging activities, significantly outperforming MSHP and ASHP, which had higher IC_50_ values. Additionally, PPHP exhibited the highest total phenolic content and Fe^2+^-chelating capacity, indicating enhanced antioxidant activity, possibly due to the enzymatic treatment and the consequent composition of peptides and phenolic compounds. Recent research has also succeeded in isolating specific antioxidant peptides from silkworm hydrolysates. For example, a 2024 study identified three novel peptides, ENIILFR, LNKDLMR, and MLIIIMR, from a pepsin-treated *B. mori* pupae hydrolysate, which showed stronger DPPH scavenging and metal-chelating activities than common synthetic antioxidants such as BHA and BHT [41]. Although we did not characterize the peptide sequences in PPHP and Alcalase has different cleavage specificity from pepsin, our findings are consistent with the notion that PPHP can be a rich source of antioxidant peptides. Further work will be required to identify the specific peptide sequences responsible for the strong antioxidant activity observed in PPHP.

The antioxidant capacity of these peptides is largely determined by their amino-acid composition and sequence. Residues such as methionine, phenylalanine, isoleucine, leucine, and lysine have been associated with DPPH radical scavenging [42], consistent with their ability to donate hydrogen atoms or participate in electron transfer [37]. In our hydrolysates, PPHP was enriched in isoleucine, leucine, and lysine, which plausibly contributed to its lowest DPPH and ABTS IC_50_ values. In contrast, phenylalanine and methionine were highest in MSHP, while tyrosine was most abundant in ASHP, indicating that not all antioxidant-linked residues are maximized in PPHP. For ABTS, high activity is often associated with cysteine, tryptophan, and tyrosine [36]; in our dataset, cystine (the disulfide form) was greatest in MSHP, tyrosine in ASHP, and tryptophan was not determined. Likewise, FRAP has been reported to correlate positively with tyrosine, methionine, phenylalanine, isoleucine, leucine, and lysine [42]. Our findings partially follow this pattern: PPHP was enriched in isoleucine, leucine, and lysine, whereas MSHP and ASHP contained more phenylalanine, methionine, and tyrosine. Taken together, the superior antioxidant responses of PPHP most likely reflect the combined effects of its branched-chain amino-acid (BCAA)-rich peptide profile and phenolic co-constituents, rather than a uniform elevation of all antioxidant-associated residues.

Our findings indicate that PPHP delivered the most robust antioxidant performance across assays (lowest DPPH and ABTS IC_50_ and highest FRAP and TPC), and that this superiority is best explained by the combined effects of its BCAA-rich peptide profile (isoleucine, leucine, and lysine) together with co-occurring phenolics, rather than by a universal elevation of all antioxidant-associated residues. The distribution of other key residues—phenylalanine and methionine predominating in MSHP, and tyrosine in ASHP—points to complementary sets of peptides and mechanisms (radical scavenging and metal interaction) across stages. On this basis, PPHP emerges as the leading candidate for downstream purification and application as an antioxidant ingredient, nonetheless, further work should define its peptide sequences, quantify peptide–phenolic interactions, address unmeasured contributors (e.g., tryptophan), and assess the bioavailability and in vivo efficacy to underpin the formulation of silkworm-derived functional foods.

## 5. Conclusions

This study demonstrates that silkworm hydrolysates from three developmental stages (MSHP, ASHP, and PPHP) are effective functional protein sources. Using a Alcalase enzyme, all hydrolysates achieved high protein yields (61–64%). The hydrolysis process significantly increased amino-acid availability, and PPHP was enriched in lysine, leucine, and histidine, whereas MSHP contained more phenylalanine, indicating enhanced nutritional quality. Functionally, MSHP delivered the strongest ACE-inhibitory activity, while PPHP showed superior antioxidant capacity, consistent with its phenolic content and radical-scavenging behavior. Collectively, these findings position Alcalase-generated silkworm hydrolysates as promising ingredients for functional foods and nutraceuticals that can provide both antihypertensive and antioxidant benefits. Future work will include peptide identification and/or in silico analyses to isolate and characterize the active peptides, define peptide–phenolic interactions, and evaluate bioavailability and in vivo efficacy to support translation into therapeutic nutrition.

## Figures and Tables

**Figure 1 foods-14-04018-f001:**
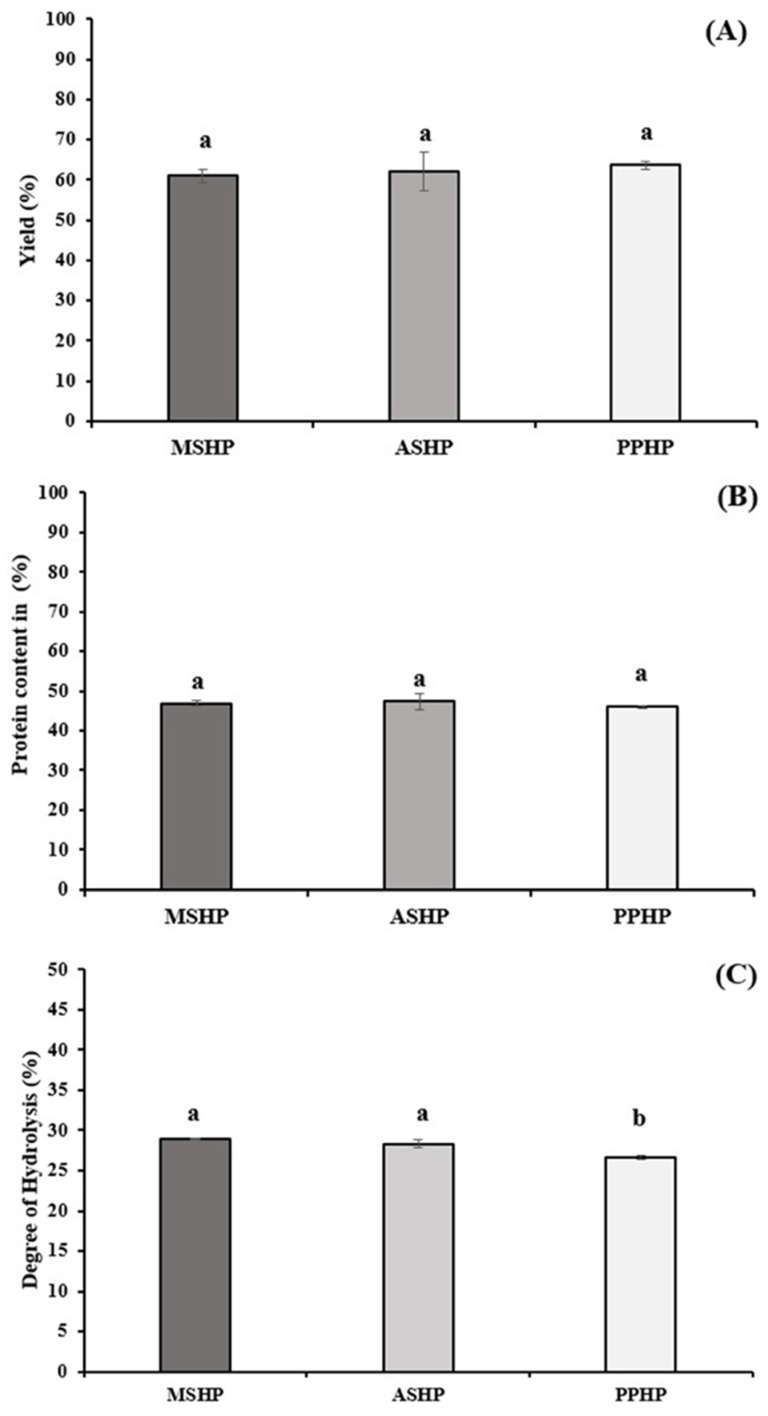
Yield (**A**), protein content (**B**), and degree of hydrolysis (**C**) of different stages of Thai silkworm hydrolysed via Alcalase^®^ 2.4 L. Data are presented as the mean ± standard deviation (SD) from triplicates. ^a,b^ Values of samples bearing different superscript lowercase letters are significantly different (*p* < 0.05). Abbreviations: MSHP: Mature Silkworm Hydrolysate powder, ASHP: A-silking silkworm Hydrolysate Powder, and PPHP: Pupae Hydrolysate Powder.

**Figure 2 foods-14-04018-f002:**
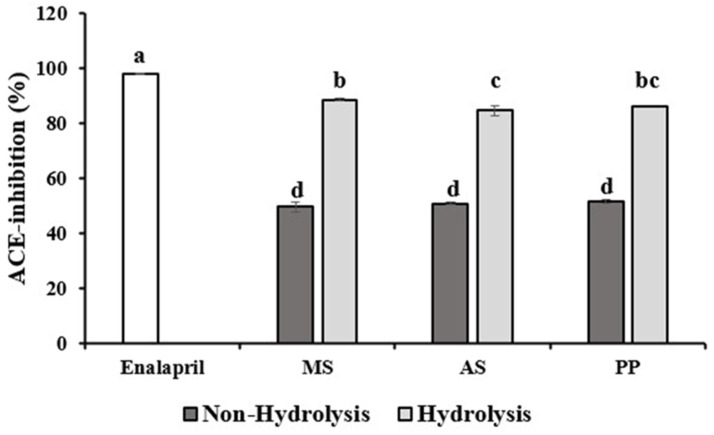
Angiotensin converting enzyme inhibition of different stages of Thai silkworm powder and Thai silkworm hydrolysate compared to Enalapril. Data are presented as the mean ± standard deviation (SD) from triplicates. Statistical analysis was conducted using one-way ANOVA followed by Duncan’s multiple range test (*p* < 0.05). ^a–d^ Values of samples bearing different superscript lowercase letters are significantly different (*p* < 0.05). Abbreviations: MS = Mature silkworm, AS = A-silking silkworm, PP = Pupae.

**Figure 3 foods-14-04018-f003:**
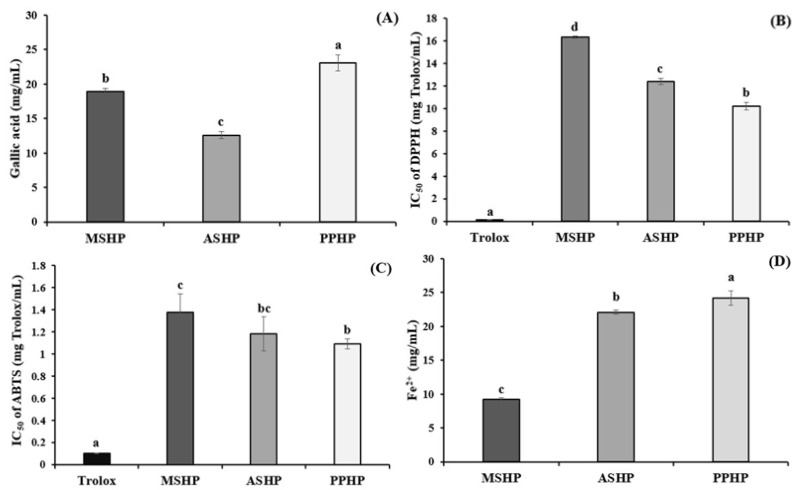
Total phenolic compounds and antioxidant activities of silkworm hydrolysate powder from different stages of silkworm. Data are presented as the mean ± standard deviation (SD) from triplicates. Statistical analysis was conducted using one-way ANOVA followed by Duncan’s multiple range test (*p* < 0.05). ^a–d^ Values of samples bearing different superscript lowercase letters are significantly different (*p* < 0.05). MSHP = Mature silkworm hydrolysate powder, ASHP: A-silking silkworm Hydrolysate Powder, PPHP: Pupae Hydrolysate Powder.

**Table 1 foods-14-04018-t001:** Chemical compositions of mature silkworm powder, A-silking silkworm powder, and pupae powder.

Properties	Type of Silkworm		
	MSP	ASP	PPP
Moisture (%)	2.51 ^a^ ± 0.24	1.54 ^b^ ± 0.06	1.77 ^b^ ± 0.16
Protein (%)	70.20 ^b^ ± 0.52	72.13 ^a^ ± 0.70	56.70 ^c^ ± 1.13
Fat (%)	17.81 ^c^ ± 0.19	21.10 ^b^ ± 0.44	30.44 ^a^ ± 0.89
Ash (%)	4.14 ^b^ ± 0.04	3.65 ^c^ ± 0.12	5.06 ^a^ ± 0.02
Carbohydrate (%)	5.34 ^a^ ± 0.38	1.50 ^b^ ± 0.96	6.03 ^a^ ± 1.50

Note a–c: Mean comparison via Duncan’s new multiple rank test. Different letters in the same row have significant difference (*p* < 0.05). MSP = Mature silkworm powder, ASP = A-silking silkworm powder, PPP = Pupae powder.

**Table 2 foods-14-04018-t002:** Amino acid profile of different stages of silkworm powder and silkworm hydrolysate powder.

Amino Acids (mg/100 g)	MSP	MSHP	ASP	ASHP	PPP	PPHP
Histidine *	290.38 ^bB^ ± 2.25	873.54 ^cA^ ± 3.92	240.77 ^cB^ ± 0.13	900.45 ^bA^ ± 0.05	319.69 ^aB^ ± 1.57	976.75 ^aA^ ± 0.71
Isoleucine *	25.95 ^cB^ ± 0.22	41.41 ^cA^± 0.08	33.65 ^bB^ ± 0.23	56.07 ^bA^ ± 0.14	51.36 ^aB^ ± 0.00	77.36 ^aA^ ± 0.00
Leucine *	50.66 ^cB^ ± 0.43	94.45 ^bA^ ± 0.75	52.29 ^bB^ ± 0.22	89.24 ^cA^ ± 0.18	61.80 ^aB^ ± 0.19	152.28 ^aA^ ± 0.12
Lysine *	101.05 ^cB^ ± 0.04	104.04 ^cA^ ± 0.06	113.17 ^bB^ ± 0.31	164.07 ^bA^ ± 0.32	142.94 ^aB^ ± 0.36	185.94 ^aA^ ± 0.84
Methionine *	27.51 ^aB^ ± 0.30	64.90 ^aA^ ± 0.09	14.59 ^cB^ ± 0.12	46.15 ^cA^ ± 0.20	20.40 ^bB^ ± 0.01	55.13 ^bA^ ± 0.12
Phenylalanine *	363.08 ^aB^ ± 8.48	1234.61 ^aA^ ± 3.27	300.50 ^bB^ ± 0.17	844.37 ^cA^ ± 3.68	113.27 ^cB^ ± 0.62	962.40 ^bA^ ± 0.83
Threonine *	72.24 ^cB^ ± 1.15	80.43 ^cA^ ± 0.04	92.32 ^aB^ ± 0.49	129.14 ^aA^ ± 0.14	76.96 ^bB^ ± 0.11	115.83 ^bA^ ± 0.60
Arginine *	52.67 ^bB^ ± 4.79	59.11 ^cA^ ± 0.49	56.70 ^bB^ ± 0.06	81.61 ^bA^ ± 0.62	68.25 ^aB^ ± 1.12	95.54 ^aA^ ± 0.35
Valine *	65.88 ^cB^ ± 0.76	119.11 ^cA^ ± 0.31	95.48 ^aB^ ± 0.08	174.82 ^aA^ ± 0.38	88.00 ^bB^ ± 0.12	137.90 ^bA^ ± 0.25
Alanine	150.44 ^aB^ ± 0.51	225.18 ^aA^ ± 0.75	136.06 ^bB^ ± 0.27	191.00 ^bA^ ± 0.35	72.83 ^cB^ ± 0.26	104.20 ^cA^ ± 0.46
Aspartic acid	5.74 ^bA^ ± 0.14	2.27 ^cB^ ± 0.06	6.73 ^aB^ ± 0.01	38.34 ^aA^ ± 0.14	3.30 ^cB^ ± 0.03	8.30 ^bA^ ± 0.11
Cystine	59.23 ^bB^ ± 2.68	181.03 ^aA^ ± 0.95	55.55 ^aB^ ± 0.56	143.73 ^bA^ ± 0.08	23.49 ^bB^ ± 0.32	118.42 ^cA^ ± 1.93
Glutamic acid	320.72 ^aB^ ± 1.39	489.45 ^aA^ ± 2.49	165.89 ^bB^ ± 0.32	271.40 ^bA^ ± 0.72	166.05 ^bB^ ± 0.16	239.78 ^cA^ ± 1.21
Glycine	125.24 ^abB^ ± 2.43	150.02 ^bA^ ± 0.02	131.58 ^aB^ ± 0.02	157.20 ^aA^ ± 0.25	117.87 ^bB^ ± 0.17	143.43 ^cA^ ± 0.28
Proline	22.15 ^cB^ ± 0.66	29.51 ^cA^ ± 0.13	40.65 ^bB^ ± 0.07	55.57 ^bA^ ± 0.51	80.91 ^aB^ ± 0.10	94.16 ^aA^ ± 0.29
Serine	63.32 ^cB^ ± 1.36	96.69 ^bA^ ± 0.70	183.97 ^aB^ ± 0.02	300.00 ^aA^ ± 0.79	68.11 ^bB^ ± 0.50	98.05 ^bA^ ± 0.51
Tyrosine	56.33 ^cA^ ± 0.83	48.22 ^cB^ ± 0.47	171.22 ^aB^ ± 8.78	246.49 ^aA^ ± 0.57	120.35 ^bB^ ± 4.61	190.78 ^bA^ ± 2.85
Total amino acids	1852.59	3839.97	1891.12	3889.65	1595.58	3756.25

Cysteine, asparagine, glutamine, and tryptophan were not reported because they could not be reliably quantified under the analytical conditions used, and standards were not available. Abbreviation: MSP = Mature silkworm powder, ASP = A-silking silkworm powder, PPP = Pupae powder, MSHP = Mature silkworm hydrolysate powder, ASHP = A-silking silkworm hydrolysate powder, PPHP = Pupae hydrolysate powder. * Essential amino acid. Within each row, lower-case superscripts compare MSP, ASP and PPP (powders), and separately, MSHP, ASHP and PPHP (hydrolysates) using one-way ANOVA followed by Duncan’s multiple range test; different lower-case letters indicate significant differences (*p* < 0.05). Upper-case superscripts compare each powder with its corresponding hydrolysate.

## Data Availability

The original contributions presented in the study are included in the article/Appendix A, further inquiries can be directed to the corresponding author.

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
