# Peer review of "The Nutritional, ACE Inhibition, and Antioxidant Properties of Hydrolysate Powders Derived from Different Stages of Thai Silkworm (Bombyx mori)"

_foods, 2025, doi:10.3390/foods14234018_

Round 1
Reviewer 1 Report
Comments and Suggestions for Authors
This study evaluated the bioactive potential of Thai silkworms at three developmental stages as sources of bioactive hydrolysates that exhibit ACE inhibitory and antioxidant activities. The study is simple as it only analysed composition and in vitro bioactivities of the samples. Identification of peptides potentially responsible for bioactivities or in silico predictions have not been carried out, which could greatly improve the work. Furthermore, the manuscript must be carefully reviewed in style and writing to avoid repetitions and redundancies and to ensure greater clarity and conciseness. Some sentences contain stylistic and technical accuracy issues that should be corrected to make the text sound more natural and accurate in scientific English. The following are some examples:
Line 76: However, a key research gap remains, as few studies …
Lines 99-112: The Thai silkworm (Bombyx mori L.), a hybrid variety reared by the Queen Sirikit Department of Sericulture, Ministry of Agriculture and Cooperatives (Thailand) was used in this study to examine three developmental stages:
- Mature silkworms (MS) were obtained by rearing silkworms until they reached the mature stage, corresponding to the final instar of larval development prior to cocoon spinning. The duration of this stage is approximately 20–25 days, depending on environmental conditions. MS are characterized by a translucent yellowish coloration.
- A-silking silkworms (AS) represents silkworms in the mature phase and typically generate cocoons naturally. However, specific techniques can stimulate silk sheet production instead of cocoons, thereby preventing metamorphosis into pupae.
- Pupae (PP) was obtained by allowing mature silkworms to spin cocoons for 3–5 days, during which they underwent metamorphosis into pupae enclosed within the cocoons.
Line 115: of silkworms (2 kg of MS, AS, and PP); Line 116: remove point after 2.5 h
Line 123: the highest ACE-inhibitory activity and yield
Lines 126-130: Briefly, 5 g of each powder were hydrolyzed with Alcalase at 5% (w/w) of the protein content in 50 mL of 0.1 mM phosphate buffer (pH 8.0), and incubated at 60 °C for 1 h. After hydrolysis, the samples were heated in a water bath at 95 °C for 10 min and centrifuged at 1,008 × g for 30 min.
Number equations consecutively.
Abbreviations should be defined the first time they appear and do not need to be repeated thereafter. Example, lines 138-139, 185, 330, 346 376, 408, …
Lines 141-162: Please rewrite to be more concise and without outline formatting
Line 163: the described method was from Nielsen et al. (2011). Moreover, several formulas are used to calculate the DH using this method, so only stating the last one is unclear. You should indicate the values ​​of α and β used, not just the value of htot. Serine is used as standard, not for preparing a standard curve.
Line 173: I assume only free amino acids (not total) are analyzed, please clarify.
Line 175: derivatization
Line 179-183: make sentences more easily readable: Mobile phase A was .. (similar for B and C)
Line 188: the diluent should be indicated
Sections 2.6, 2.7, 2.8: need to be rewritten and reviewed. They contain grammatical and typographical errors, as well as redundant or unnecessary information. The ACE inhibition equation is missing, units such as min or h should be abbreviated, ... Be careful with expressions; for example, the antioxidant methods are not originally from Anuduang et al., so you could say “as described in…” instead. Similar issues apply throughout the manuscript.
Table 2 could include the total amino acid content for greater comparison between samples.
Please correct the X-axis of the Figure 2B
Line 376: remove Furthermore; line 378: in all assays instead of all parameters
Line 381; Figure 3D: why FRAP results were not expressed as IC50 (mg/mL)?
Line 403: complete according to… ; Line 404: It has already been indicated before.
Section 4.2: Are results given for previous studies based on free or total amino acids? You should consider this to compare and discuss with your results, especially in those samples before hydrolysis.
Line 463: This sentence is confusing. Is protein lost or degraded into peptides and free amino acids due to enzymatic action?
Line 476: what does “under comparable conditions” mean?; lines 479-480: The sentences are not connected and lack coherence.
Line 495: it should be: smaller peptides with potential functionality or bioactivity. Lines 499-503: This should be indicated later, because the bioactivity results have not yet been discussed.
Section 4.4: MW fractionation of sample MSHP was not previously indicated in Material and methods or Results sections, please add that information. Rewrite the first sentence as: MSHP, which showed the highest ACE-inhibition, was fractionated into three molecular weight ranges (≥10, 3–10, and ≤3 kDa) to clarify the molecular size distribution of ACE-inhibitory peptides. What was the concentration of the fractions tested for ACE inhibition? In my opinion, supplementary material could be included in the manuscript as a single figure with 2 X-axes (ACE inhibition and yield) for better discussion.
Lines 516-524: is this information actually specific to insect-derived peptides, or does it refer to ACE inhibitors in general, which could include those from insects?
Lines 551-552: Why only Trp-Trp is indicated as abbreviated form? Please unify.
In my opinion, section 4.4 is a long section, with repeated and redundant information, it could be organized and simplified for getting a better discussion.
Line 604: “but also yields bioactive peptides with specialized functions” , what does it mean?
Line 610: What does “robust” mean? And following sentence should be: possibly due to the enzymatic treatment and the consequent composition of peptides and phenolic compounds.
Lines 616-617: There are many peptide sequences responsible for strong antioxidant activity; therefore, it is not possible to determine whether PSPH might contain similar peptides, and especially since pepsin and Alcalase have different cleavage sites.
Line 632: BCAA was not defined previously. Similar to 4.4, avoid unnecessary repetition and redundant restatements in section 4.5
Author Response
Dear Reviewer 1,
Thank you very much for your valuable comments and suggestions. I have provided detailed responses to all of your points in the attached file.
Kind regards,

Reviewer 2 Report
Comments and Suggestions for Authors
An interesting manuscript focusing on nutritional, ACE inhibition, and antioxidant properties of silkworm hydrolysates. I have no comments on the Introduction, Discussion, and Conclusion chapters. I have the following comments on the Material and Methods and Results chapters:
L 116: Fill in the information about how long the silkworms were boiled.
L 143: I would replace “determined” by “calculated”.
L 151: Add the specification of petroleum ether (boiling point range, purity, producer).
L 160: Why did you use the nitrogen-to protein conversion factor 6.25? There is a specific Kp = 5.24 for Bombyx mori (Ochiai et al. 2024).
L 161: Your calculation of carbohydrates does not include chitin, because this polysaccharide contains nitrogen and is therefore determined via Kjeldahl and included into crude protein.
L 173: It would be good to underline in the whole text that you analysed free amino acids (and not all amino acids including those in proteins and peptides).
L 205: The equation is missing.
L 209: “mg” or “mL”?
L 214: Which concentration range of gallic acid standards for calibration was prepared?
L 240: Correct “40 mM” and “20 mM” to “40mM” and “20mM” in case of abbreviations.
Table 2: Add information why cysteine, asparagine, glutamine, and tryptophane are not included into the table.
Figure 2 shows and compares also the results between non-hydrolysed and hydrolysed silkworm powders. Could you please add the data about non-hydrolysed samples to Figure 1 and Figure 3 enabling also comparison between non-hydrolysed and hydrolysed samples?
L 552, 600, and 644: “in vitro” and “in vivo” are the Latin words and should be typed in Italics.
Author Response
Dear Reviewer 2,
Thank you very much for your valuable comments and suggestions. I have provided detailed responses to all of your points in the attached file.
Kind regards,
